# PREDICTING THE ENCODING ERROR
# OF IMPLICIT NEURAL REPRESENTATIONS

## ABSTRACT

Implicit Neural Representations (INRs), which encode signals such as images, videos, and 3D shapes in the weights of neural networks, are becoming increasingly popular. Among their many applications is signal compression, for which there is great interest in achieving the highest possible fidelity to the original signal subject to constraints such as neural network size, training (encoding) and inference (decoding) time. But training INRs can be a computationally expensive process, making it challenging to determine the best possible tradeoff under such constraints. Towards this goal, we propose a novel problem: predicting the encoding error (training loss) that an INR will reach on a given training signal. We present a method which predicts the encoding loss that a popular INR network (SIREN) will reach, given its network hyperparameters and the signal to encode. Our predictive method demonstrates the feasibility of this regression problem, and allows users to anticipate the encoding error that a SIREN network will reach in milliseconds instead of minutes or longer. We also provide insights into the behavior of SIREN networks, such as why narrow SIRENs can have very high random variation in encoding loss, and how the performance of SIRENs relates to JPEG compression.

## 1 INTRODUCTION

Since the introduction of neural radiance fields in 2020, there has been a swell of interest in neural networks as representations of data. Such networks, commonly called *implicit neural representations* (INRs), or *coordinate networks*, are trained to fit relatively low-dimensional signals, such as 2D images, 3D shapes (typically as occupancy functions or signed distance functions), or neural radiance fields (NeRFs). Implicit neural representations have proven useful for a diverse range of tasks, including super-resolution (Chen et al., 2021b), novel view synthesis (Park et al., 2021), video manipulation (Pumarola et al., 2021; Gao et al., 2021; Mai & Liu, 2022; Li et al., 2021; Xian et al., 2021; Tretschk et al., 2021), and compression.

There is great interest in achieving the highest possible fidelity to the encoded signal subject to constraints such as network size, training and inference time. In the context of signal compression, researchers wish to minimize the network's size while maintaining a low encoding error.

In this paper, we study SIREN networks trained to fit natural images. Multilayer perceptrons with periodic activation functions, or SIRENs, have many advantages. They are simple to implement and quick to train. They train stably even when 10+ layers deep, and have well-behaved derivatives (Sitzmann et al., 2020). Therefore, they have become a baseline architecture for INRs, for example, see Dupont et al. (2021); Martel et al. (2021); Lindell et al. (2022); Chen et al. (2021a). Meanwhile, natural images have become a standard training target for the evaluation of coordinate network architectures, as seen in papers such as Sitzmann et al. (2020), Tancik et al. (2020), and Dupont et al. (2021).

In the context of images, by far the most common metric by which SIRENs are judged is mean squared error (MSE), often formulated as peak signal-to-noise ratio (PSNR). Practitioners often seek to maximise their INRs' PSNR subject to constraints such as training time (for signal compression, this is encoding time), number of trainable parameters (for compression), or forward-pass computation time (for decoding efficiency). However, training SIRENs can be time-consuming. For example, training a SIREN to compress a 512x768 pixel image to 0.3 bits per pixel with the method

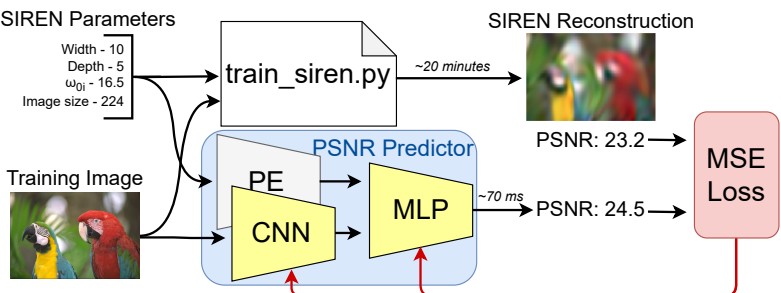

Figure 1: Overview of our many-architecture encoding error predictor. First, we train SIRENs on many images with a random sampling of hyperparameters. Then, our many-architecture predictor takes the same inputs as the SIREN training script, and predicts the SIRENs' PSNR. The training image is passed through a Convolutional-network-based feature extractor (CNN), while the SIREN hyperparameters are fed through a positional encoding (PE). Then both are concatenated and passed through a fully-connected network (MLP) which predicts what PSNR the training script will reach.

from Dupont et al. (2021) takes 50 minutes on a Titan X GPU. Suppose, that we wish to compress an image using the smallest SIREN that will achieve a PSNR of at least 30 dB. Without a predictive model of SIREN performance, we will have to perform a time-consuming hyperparameter sweep to determine the best SIREN architecture for our task. A predictive model of what encoding error a given SIREN will reach will allow us to make quick, informed choices across a broad range of applications.

To address this issue, we develop a predictive model of how well a SIREN will encode a given image. Our model takes in a set of SIREN hyperparameters and the target image, and predicts the encoding error, measured as PSNR, that the SIREN will achieve. Within the training domain, our model makes accurate predictions of final training loss in just milliseconds, as opposed to minutes of training.

To train this model, we develop a dataset of 300,000 SIRENs, trained across 100,000 photographs and a range of SIREN hyperparameters. This dataset contains the hyperparameters, training loss curves, and final weights of each SIREN. Initial weights for each SIREN can be reconstructed using our source code.

Along the way, we offer some insights about the behavior of SIRENs. We observe that:

- Deep, narrow SIRENs like those used by Dupont et al. (2021) have significant random variation in their encoding error, which is largely explained by the initialization of the first layer.
- SIRENs approximately follow scaling power-laws as described in Bahri et al. (2021).
- There is a strong correlation between JPEG and COIN compression.

## 2 RELATED WORK

### 2.1 A BRIEF OVERVIEW OF THE SIREN ARCHITECTURE

SIRENs, introduced by Sitzmann et al. (2020), are simply multilayer perceptrons with sinusoidal activations. A SIREN $\Phi(x)$ has the following form:

$$
\begin{aligned}
\Phi(x) &= W_L h_{L-1}(x) + b_L \\
h_l(x) &= \sin(W_l h_{l-1}(x) + b_l) \quad \text{for } l = 1, \ldots, L-1 \\
h_0(x) &= \omega_0 x
\end{aligned}
\tag{1}
$$

where $L$ is the number of layers, $W_0, W_1, \ldots, W_L$ and $b_0, b_1, \ldots, b_L$ are the network's weight matrices and bias vectors, respectively, and $\omega_0$ is a scalar hyperparameter which controls the spatial frequency of activations in the first hidden layer $h_1(x)$, and hence the spectral bias of the network.

## 2.2 COMPRESSION WITH IMPLICIT NEURAL REPRESENTATIONS

INR-based data compression is the application most directly relevant to our research. Since the introduction of the SIREN network by Sitzmann et al. (2020), researchers have sought to use SIRENs as compressed representations for images. In 2021 Dupont et al. (2021) showed that SIRENs can outperform JPEG compression at low bitrates. We use their architecture and training setup as a baseline for our experiments.

Since then, several schemes have been devised to improve a SIREN's compression capabilities. These schemes employ a combination of meta-learning (Dupont et al., 2022b; Schwarz & Teh, 2022; Lee et al., 2021; Strümpler et al., 2022), weight pruning (Lee et al., 2021; Ramirez & Gallego-Posada, 2022), and quantization (Gordon et al., 2023) to improve the rate-distortion curves of SIREN-based image compression. Guo et al. (2023) extend this compression paradigm to Bayesian neural networks. All of these projects attempt to obtain the best-possible rate-distortion curves for some set of images, although the size and content of the images studied varies across papers. Gao et al. (2023) propose a new loss term to enforce structural consistency between the reconstructed and ground truth images based on their segmentation maps. Almost all of these works build upon the SIREN architecture; therefore we believe that our observations about SIREN-based image compression are broadly relevant to this whole body of work.

Dupont et al. (2022b) shows that INRs can be used to compress many other data types, including sound, MRI, and weather data. Video is an especially common and data-intensive modality which has received special attention: while Sitzmann et al. (2020) and subsequent papers such as Mehta et al. (2021) applied SIRENs to relatively small (e.g. 7x224x224) videos, more specialized INR architectures such as NeRV (Chen et al., 2021a), E-NERV Li et al. (2022), NVP Kim et al. (2022), and many others (Rho et al., 2022; Zhang et al., 2022; He et al., 2023; Chen et al., 2022; Maiya et al., 2023), have reached performance comparable with conventional video compression methods such as H.264 Wiegand et al. (2003) and HEVC Sullivan et al. (2012). While outside the scope of the current work, our encoding error prediction method could be extended to these other modalities.

## 2.3 THEORETICAL MODELS OF NEURAL NETWORKS

Our primary goal is to develop a predictive model of how well a SIREN can encode a target image. While our methods are data-driven and empirical, others works have developed relevant theoretical models of network training dynamics, which may be predictive of SIREN performance. In particular, Neural Tangent Kernel (NTK) theory (Jacot et al., 2018) offers one possible tool for modelling the performance of SIRENs. As networks become sufficiently overparameterized, they enter a *lazy training regime*, where their training is well-approximated by Taylor series expansion around their initialization (Atanasov et al., 2023). Even outside of this lazy training regime, researchers have successfully applied NTK theory to explain certain properties of coordinate networks. For example, Tancik et al. (2020) use NTK theory to explain the advantage conferred by using a random Fourier feature (RFF) positional encoding for coordinate networks. As they show, applying NTK theory to neural networks with an RFF positional encoding naturally leads to a frequency-based perspective of the network's behavior. Ronen et al. (2019) perform a more extensive analysis of the convergence rate of neural networks to functions of different frequencies, and confirm their theory matches observations for a few circumstances. Yüce et al. (2022) employ a different mathematical approach to reach similar conclusions about the spectral bias of the fourier-feature networks and SIRENs.

However, none of these works can precisely predict the loss curves of *underparameterized* networks which lack enough trainable paramters to fully fit their training data. While some general arguments from NTK theory still apply, the approximation is too rough to make the kind of fine-grained PSNR predictions we seek.

For example, in *Predicting Training Time Without Training*, Zancato et al. (2020) use neural tangent kernels to predict the fine-tuning loss curves of pretrained image classification networks. On the surface, their problem is extremely similar to ours. But their networks operate in the lazy training regime, while ours do not. Due to this difference, we found that their method does not transfer well to our setting. See Appendix A.4 for more details.

Bahri et al. (2021) propose a theoretical model which is more applicable to our setting. They identify four regimes of neural network training dynamics, including a *resolution-limited regime*, where the

number of training samples far exceeds the number of parameters. They argue that the model's loss will decrease as a power law of the number of trainable parameters. In Section 6.1, we show that this model is in closer accordance with our observations. However, this power law only tells us how the loss will vary with model size, with everything else held constant. We are specifically interested in how PSNR varies with the *training data*, and here their model offers no guidance.

## 2.4 META-LEARNING IMPLICIT NEURAL REPRESENTATIONS

Meta-learning, such as MAML Finn et al. (2017), optimizes the initial weights of a neural network to converge quickly to any one of a set of target functions. Meta-learning can dramatically reduce the amount of time needed to fit a SIREN to a given signal (Tancik et al., 2021; Dupont et al., 2022a; Lee et al., 2021). Because meta-learning optimizes the initial weights of the network, it implicitly holds the network's hyperparameters, such as its width and depth, fixed. However, our encoding error prediction model works over a continuous range of possible hyperparameters.

## 3 DATASETS

In this paper, we make use of two image datasets: Kodak (1991) and MSCOCO (Lin et al., 2014). The Kodak dataset is a popular reference dataset in image compression literature, and we use it for our small-scale exploratory studies in Sections 4, 5, and 6.1. But the Kodak dataset consists of only 24 images, and our predictive models of SIREN encoding error require larger datasets for training and evaluation. Therefore, we train 300,000 SIRENs on a set of 100,000 images from the the MSCOCO dataset, all downsampled and center-cropped to 512x512 pixels.

For each of these 300,000 SIRENs, we record their hyperparameters, training loss curves, and final weights after 20,000 steps of training. These 300,000 SIRENs are divided into two datasets. 100,000 of these SIRENs all share the same architecture, and only vary by the image that they encode. We call this collection of SIRENs the *single-architecture dataset*. The remaining 200,000 comprise the *many-architecture dataset*, and they are trained across a range of architectural hyperparameters. In total, these 300,000 SIRENs took roughly 12,000 GPU hours to train. The rest of Section 3 describes these two sets of SIRENs in more detail.

## 3.1 THE SINGLE-ARCHITECTURE DATASET

We train 100,000 SIRENs on the 100,000 images in our random sampling of the MSCOCO dataset. Each image is downsampled to 224x224 pixels and then used to train a SIREN with 8 layers, 32 hidden units per layer, and a $\omega_0$ of 15. If we store the SIREN parameters at 16-bit precision, this corresponds to a 0.9 bpp representation of each image. The PSNRs of the resulting images have a mean of 30.53 dB, a standard deviation of 4.41 dB, and are precisely Gamma-distributed.

## 3.2 THE MANY-ARCHITECTURE DATASET

We also train a collection of SIRENs over a range of hyperparameter values. For this group of SIRENs, we select 200,000 random samples of training parameters from the following range of options:

- **Target image** - All images were chosen from our subset of **100,000 photographs** randomly sampled from the MSCOCO dataset. The training/validation/tests splits are done such that no image appears in two splits.

- **Image size** - We train our SIRENs on square images **between 112 and 512 pixels** in width.

- **Network Depth** - Following Dupont et al. (2021), we sweep through networks **between 2 and 12 layers deep**. We notice a dropoff in performance at both ends of this distribution, suggesting we have swept the useful range.

- **Network width** - All SIRENs we train have a constant "width," i.e. a constant number of hidden units per layer. We sample the number of hidden units such that the SIRENs' compression ratios follow a log-uniform distribution **between 0.5 and 9 bits per pixel** (bpp).

- $\omega_0$ - This is an important SIREN hyperparameter which determines the spatial frequency of the activations which come out of the first layer. Sitzmann et al. (2020) found that a $\omega_0$ of 30 worked well for their applications. We find, perhaps unsurprisingly, that the optimal choice of $\omega_0$ varies with both the image size and image content. In general, larger images with more high frequency detail benefit from a higher $\omega_0$. Therefore, we randomly sample a value $\gamma = \frac{\omega_0}{image\ size}$ from a log-uniform distribution between 0.02 and 0.12, and set $\omega_0 = \gamma \times image\ size$. Like the number of layers, this distribution appears to span the optimal range of $w_0 i$ values.

Two important parameters which we do not vary in this paper are 1) number of training steps and 2) learning rate. We find that across all of our networks, a learning rate of 0.001 is near-optimal. We also find that our SIREN networks show diminishing returns in PSNR after around 20,000 training steps. Indeed, during the last 1,000 steps, the median SIREN in our dataset improved by only 0.0016 dB PSNR, and the 99th percentile network improved by 0.070 dB. Therefore, we fix the number of training steps to 20,000 for all our networks.

Because this random sampling includes many suboptimal architecture choices, the PSNR of the many-architecture dataset tends lower than in the single-architecture dataset. The average PSNR of this group is 28.12 dB, the standard deviation is 4.86 dB. Figure 10 in the Appendix shows the gamma-distributed histograms of PSNR values for the single- and many-architecture datasets.

## 4 RANDOM VARIATION IN SIREN ENCODING ERROR

We find that there is significant random variation in what encoding error a given SIREN will reach, based solely on its random initialization. We quantify this random variation with an estimate of the standard deviation in PSNR among SIRENs which have been trained identically except for their random initialization. For the 10-layer-deep, 28-neuron-wide SIRENs trained on the Kodak dataset from Dupont et al. (2021), the random variation has of approximately $\pm\,0.18$ dB.

The magnitude of this random variation decreases with network width, and can be largely attributed to the random initialization of the first layer. The consequence is that very narrow, deep SIRENs like the ones used by COIN are the most affected, but this effect can be ameliorated by holding the first layer fixed, or by using a meta-learned initialization. (See Appendix A.5 for further explanation.)

This random variation is important to keep in mind when comparing the performance of different SIREN architectures and training schemes. For example, the original COIN paper conducted a small-scale hyperparameter sweep suggesting that 10 layers was an optimal depth for their networks (Dupont et al., 2021). This result was obtained by training one network at each depth, and selecting the network with the best PSNR. We retrain these networks 10 times each with different random seeds, and then bootstrap a random sample of possible experimental outcomes which could have occurred. We found that 95% of the time, their procedure selects optimal depths between 7 and 13 layers deep. This demonstrates that random variation can play an important role in the outcome of such small-scale experiments.

This random variation is also important to keep in mind when evaluating our encoding error prediction models, because it is a source of irreducible error in our prediction problem. If PSNR varies randomly with the network's random initialization by, say, 0.3 dB, we cannot possibly predict the encoding error more accurately than that. Random variation places an upper-bound on how accurate our encoding error prediction models can be.

Therefore, we estimate the random variation in encoding error in the single- and many-architecture dataset. To do so, we randomly sample 10,000 SIRENs from each of those datasets, and retrain them with different random seeds. We estimate the aggregate standard deviation (or RMSE) of random variation using the formula:

$$RMSE = \sqrt{\frac{1}{2N}\sum_{i=1}^{N}(y_{i,1} - y_{i,2})^2} \qquad (2)$$

where $N = 10,000$ is the number of pairs of SIRENs which differ only by their random seed, and $y_{i,1}$ and $y_{i,2}$ are the PSNRs of the $i$th pair of SIRENs. (See Appendix A.5.1 for an explanation of

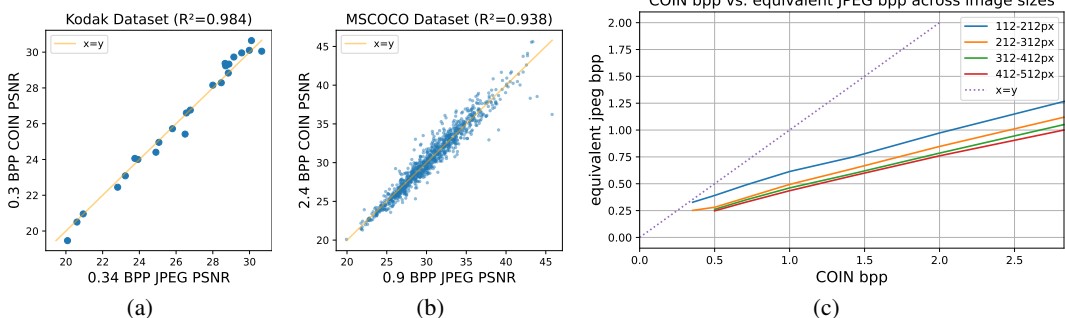

Figure 2: Relationship between SIREN and JPEG representations. In (a) and (b), each data point represents an image, and the PSNR to which it can be compressed by JPEG and COIN respectively. In (c) we show which JPEG compression ratio reaches the same PSNR as COIN for a given image size and COIN compression ratio.

this formula). In Section 6, we will refer to this statistic as the *irreducible error* of a given dataset, because it sets a lower bound on the error our predictive model can reach.

## 5 COMPARING COIN AND JPEG COMPRESSION

There is a strong correlation between how well JPEG and COIN compression can fit a given image. This suggests a simple technique for predicting SIREN encoding error: find the encoding error of a SIREN architecture on a few images, and then use a standard zeroth-order numerical optimizer (e.g. scipy.optimize) to find the corresponding JPEG compression ratio which maximises the correlation between JPEG and SIREN encoding error.

Using the Kodak dataset and SIREN hyperparameters from Dupont et al. (2021), we train a SIREN on each image to obtain a 0.3 bits-per-pixel (bpp) compressed representation of each image. Using JPEG2000 and compressing to 0.34 bpp, we find that the JPEG's PSNR explains 98% of the variance in SIREN's PSNR (See Figure 2a).

This correlation is not quite as strong for our single-architecture SIREN dataset. Compressing these images to 0.9 bpp using JPEG2000 explains 94% of the variance in SIREN PSNR. JPEG PSNR predicts the SIRENs' PSNR to within 1.0 RMSE. See Figure 2b.

Comparing JPEG and SIREN compression for our many-architecture dataset is trickier. For the Kodak and single-architecture sets of SIRENs, we treated JPEG compression as a one-parameter predictive model of SIREN PSNR. But with the multiple-architecture dataset, each SIREN architecture is best predicted by a different JPEG compression ratio. Without a technique for choosing a JPEG compression ratio for each possible SIREN architecture, we cannot use JPEG compression to predict SIREN performance across a range of architectures. We do not go so far as to provide such a technique.

Instead, for each trained SIREN network in our dataset, we find the JPEG compression ratio that reaches the same PSNR, and then analyze the relationship between COIN bpp and equivalent JPEG bpp. For each image size, we observe a linear relationship between the compression ratio of our optimal-hyperparameter SIREN networks (as measured in bits per pixel) and the equivalent JPEG compression ratio, see Figure 2c. Our single-architecture dataset mostly contains SIRENs at bpps which are not competitive with JPEG, but notice that extrapolating the lines to the left indicates that our SIRENs should outperform JPEG at between 0.1 and 0.3 bpp. This is consistent with Figure 2 of (Dupont et al., 2021), which shows COIN outperforming JPEG compression below 0.3 bpp.

## 6 PREDICTING SIREN ENCODING ERROR

Can we predict the encoding error a given SIREN will reach on a given target signal? Section 5 showed that we can predict SIREN PSNR from JPEG compression PSNR with over $94\%$ explained variance. Section 6.1 demonstrates a power-law relationship between SIREN width and encoding error, offering a partial answer to the prediction problem. Section 6.2 explores a simple method for predicting SIREN encoding error by only training the SIREN for a small number of iterations, and then extrapolating. In Sections 6.3 and 6.4, we train neural networks to predict the encoding error that SIRENs in our Single- and Multiple-Architecture datasets will reach.

We report the accuracy of our encoding error predictors with two related metrics: root-mean-squared-error (RMSE) and explained variance (EV). For all our results, explained variance is equivalent to the coefficient of determination, $R^2$. For example, when we say that some predictive model explains 95% of the variance in encoding error, this means that the $R^2$ coefficient between the predicted and actual encoding error is 0.95.

### 6.1 SIREN SCALING FOLLOWS POWER LAWS

Bahri et al. (2021) present a simple theoretical model of neural networks trained with large amounts of data and limited parameters, which they refer to as the *resolution-limited* regime. According to their model, MSE loss $L$ should decrease with the number of parameters $P$ according to a power law $L(P) = \Omega(P^{-2/d})$, where $d$ is the implicit dimension of the data manifold.

Since PSNR is proportional to the log of MSE, this law implies that the PSNR should grow linearly with $\log(P)$. Figure 3 shows PSNR vs. $\log(P)$ for SIRENs trained on 6 of the 24 Kodak images, for networks with ten layers, and widths from 4 to 128 neurons per layer. For a limited range of network widths, this linear relationship holds up quite well. As a rule of thumb, we find that each doubling of the network size improves PSNR by about 2 dB.

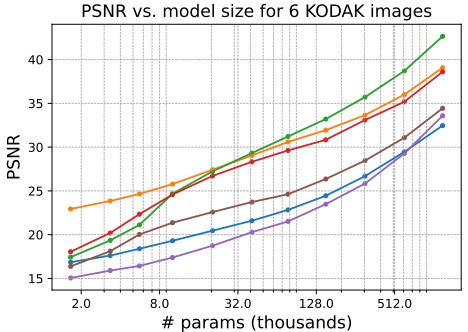

Figure 3: Scaling curves for SIRENs trained on the first six images in the Kodak dataset. Each line represents the scaling curve for a different image.

However, the exponent of this power law appears to be contingent on our experimental setup. Much smaller or larger SIRENs, or SIRENs trained on very different data, will have different scaling curves. For example, notice that the image represented by the green line follows a steeper power law exponent than the other five images. And as we zoom out to consider networks which are orders of magnitude larger or smaller than the original COIN network, the relationship between PSNR and $\log(P)$ takes on positive curvature.

These power laws allow us to make local predictions about how the encoding error will change with the SIREN width. But to predict encoding error across other variables, such as the target image content, we will need a model which depends on those inputs as well.

### 6.2 EXTRAPOLATING FROM PAST PSNR TO FUTURE PSNR

Perhaps one of the simplest methods to predict the encoding error a SIREN will reach after $n$ training steps is to train the SIREN for $m < n$ steps, observe its encoding error, and extrapolate what the error will be at training step $n$, using linear regression. For example, the COIN networks are trained on the Kodak dataset for 50,000 steps each. The PSNR of each SIREN after 10,000 training steps explains 99.7% of the variance in PSNR after 50,000 steps. Table 1 shows the explained variance (i.e. $R^2$ coefficient) you can reach with this prediction method for each of our datasets.

We would like to predict the encoding error to a high accuracy in orders of magnitude less time than it takes to train the network. Table 1 shows that this simple extrapolation method is inadequate to achieve this goal. By step 10,000, these SIRENs are approaching their asymptotic behavior of sharply diminishing returns in PSNR per training iteration. In this regime, there is a strong corre-

Table 1: Explained variance (EV) in PSNR after $n$ training steps from the PSNR after $m$ training steps, as calculated by $R^2$ score. For example, the PSNR after 1,000 steps explains 95.8% of the variance in PSNR after 50,000 steps for the SIRENs trained with the images and architecture from COIN (Dupont et al., 2021).

| Dataset | $n$ steps | EV (%) after $m$ steps ↑ | | |
|---|---|---|---|---|
| | | 100 | 1,000 | 10,000 |
| COIN | 50,000 | 2.5 | 95.8 | 99.7 |
| single-arch. | 20,000 | 25.8 | 93.0 | 99.8 |
| multi-arch. | 20,000 | 20.7 | 90.9 | 99.9 |

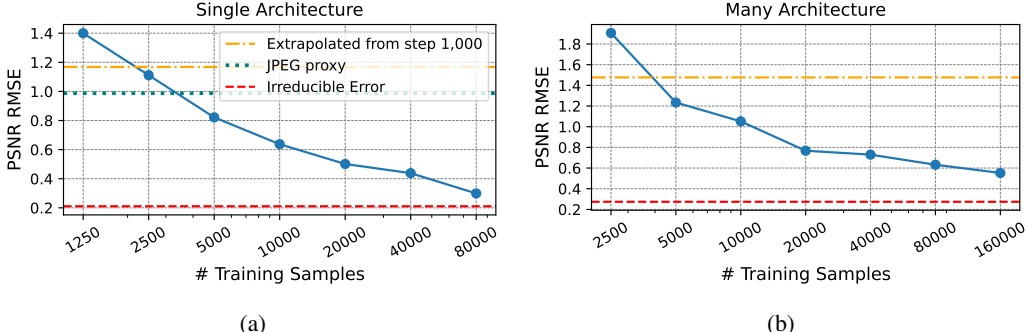

(a)                                    (b)

Figure 4: Number of training samples vs. test set error (PSNR RMSE) for the (a) single and (b) multiple architecture datasets. Horizontal lines represent 1) the RMSE we get by linearly extrapolating the PSNR from training step 1,000, as described in Section 6.2, 2) RMSE from predicting PSNR with a "JPEG proxy model" as depicted in Figure 2, and 3) the irreducible error of the regression problem, as described in Section 4.

lation with the final PSNR, but only because the PSNR doesn't change very much in the remaining training steps: From step 10,000 to step 50,000, the COIN networks only improve by an average of 0.4 PSNR. We would be better off predicting the final PSNR using an equivalent JPEG compression, as explained in Section 2. In the next section, we will see that we can do better by solving this regression problem using neural nets.

## 6.3  PREDICTING LOSS FOR A SINGLE ARCHITECTURE

For the single-architecture dataset, all aspects of training besides the target image are held constant. The only variable we use to predict the SIREN's final PSNR is the target image. We train a convolutional network to take in a target image, and predict what PSNR our SIREN will reach on that image (see Appendix A.1 for details). We chose a variant of Normalization-Free Network from Brock et al. (2021) as our backbone CNN architecture, due to its high accuracy on the validation set. Appendix A.2 describes our backbone architecture selection procedure in more detail.

Using NFNet, we predict the SIRENs' PSNR to within 0.30 dB RMSE, with an $R^2$ score of 0.996. The irreducible error of this prediction problem is approximately 0.21 dB PSNR. With a forward-pass time of 70 ms, as opposed to an original training time of 2.5 minutes per network, this allows us to predict PSNR  2,000× faster than we could by training the SIREN. Figure 4a shows how test error decreases with the size of the training set. The scaling curve is remarkably uniform, and suggests that even larger training sets would continue to improve test accuracy.

### 6.4 Predicting Loss Across Many Architectures

For the multiple-architecture dataset, we have several independent variables from which to predict PSNR: the training image, $\omega_0$, network width, and network depth. We feed all of these variables into a neural network whose architecture is depicted in Figure 1. See Appendix A.1 for details.

Using this setup, we train our model to predict the PSNR to within 0.55 RMSE, to an $R^2$ score of 0.987. A portion of the higher error comes from a higher irreducible error of 0.27 RMSE, due to the inclusion of some suboptimal SIREN architectures which have a very high random variation in PSNR. Figure 4b shows how test set error decreases with the size of the training set.

## 7 Conclusions

We have observed several important trends in SIREN encoding error. First, SIRENs have significant random variation in performance between random initializations, which is mostly explained by the randomly generated "positional encodings" of their first layer. This effect gets more pronounced as the networks become narrower. Pre-selecting a high-quality positional encoding layer instead of sampling one randomly appears to be an easy win for improving the encoding error of a narrow SIREN. We also find that SIREN PSNR correlates strongly with JPEG PSNR. SIRENs outperform JPEGs when the representation is very small: representing a small image or to a very low bpp.

We have shown that a relatively standard deep-learning architecture can solve a novel regression problem: predicting the encoding error a SIREN will reach on a given signal. Our encoding error prediction networks compare favorably to three alternative approaches: 1) using JPEG compression loss as a proxy for SIREN loss, 2) linearly extrapolating future PSNR from a few steps of gradient descent and 3) NTK-based approximations of SIRENs in Appendix A.4. We show that this works for a single SIREN architecture, and also across a range of possible architectures.

## 8 Future Work

The "holy grail" of this line of research is a theoretically well-founded, low-parameter-count model which can predict the training loss of underparameterized neural networks with high accuracy. Towards this aim, we suspect that our dataset of 300,000 SIRENs contains a lot of scientific value beyond the scope of this work. SIRENs trained to fit photographs using mean-squared-error loss are a promising "model" deep learning problem, in the same way that fruit flies are a common model organism. This model problem has several advantages:

- **Architecturally Simple**: SIRENs are a simple variation on multi layer perceptrons, which are one of the most ubiquitous architectural building blocks in deep learning.
- **Computationally Cheap**: In the context of our coordinate networks, a single photograph represents an entire training dataset. Both our SIRENs and their training data (photographs) have a low memory footprint, less than a megabyte each. Indeed it is this quality which has allowed us to publish a dataset of 300,000 of them.
- **Understandable**: Perhaps no naturally-occurring kind of dataset is more amenable to visualization than a photograph. The training samples, image pixels, are positioned on a regular grid, making them uniquely amenable to mathematical analyses such as the discrete Fourier transform. The loss function used to train our models, $L_2$ loss, is extremely well-studied and mathematically well-behaved.

Our 300,000 trained SIRENs provide a lot of empirical data about how simple neural networks learn to approximate functions. We hope that researchers who are interested in modeling the behavior of neural networks, especially networks in the *resolution-limited regime* (Bahri et al., 2021), can evaluate their theories against our data.

## 9 Reproducibility Statement

Appendix A.1 provides implementation details for our encoding error prediction models. Upon publication of this paper, we will publish our code and our dataset of 300,000 trained SIRENs from

which this work is derived. This dataset includes the SIRENs' hyperparameters, training loss curves, and final weights. Each SIREN also includes the random seeds used to initialize Numpy and PyTorch before each training run. That said, we have not confirmed that different computing environments, CUDA versions, etc. always lead to the exact same results given our initial conditions.

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

## A  APPENDIX

### A.1  ENCODING ERROR PREDICTOR IMPLEMENTATION DETAILS

#### A.1.1  SINGLE ARCHITECTURE IMPLEMENTATION DETAILS

To train our single-architecture encoding error predictor, we use the ADAM optimizer with a learning rate of 0.0001 and a batch size of 8. PSNR values are normalized to have a mean of 0 and a standard deviation of 1 for the training data. We start from a backbone pretrained classification network (See Section A.2 for details). We remove the classification head, and replace it with a "regression head", a fully-connected layer which outputs a single value: the PSNR prediction. We freeze all weights in the network except the regression head and perform 10 epochs of training, after which we unfreeze the entire network for an additional 10 epochs of training. Finally, we select network checkpoint from the epoch with the best validation set accuracy.

#### A.1.2  MANY ARCHITECTURE IMPLEMENTATION DETAILS

The many-architecture encoding error predictor is slightly more complex, and has more implementation details to consider.

First is the positional encoding, which encodes the SIREN hyperparameters (width, depth, $\omega_0$, and image size), into a vector in $\mathbb{R}^{80}$. We use the positional encoding scheme from Mildenhall et al. (2021):

$$\gamma(p) = (\sin(2^0\pi p), \sin(2^0\pi p), \ldots, \sin(2^{10}\pi p), \sin(2^{10}\pi p)) \tag{3}$$

First, the hyperparameters are normalized to lie in the range (0,1). Log-uniformly distributed parameters $w$ and $\omega_i$ are re-scaled by taking their logarithm first, before normalization. For normalized width $w$, depth $d$, image size $s$, and $\omega_0$, the full positional encoding of the hyperparameters is:

$$\Gamma(w, d, s, \omega_0) = (\gamma(w), \gamma(d), \gamma(s), \gamma(\omega_0)) \tag{4}$$

This positional encoding and the feature vector from the last layer of the classification network are concatenated and fed into the MLP regression head. In our experiments, we found that the performance of the multi-architecture classifier was not too sensitive to the width and depth of this MLP head, so we keep the network relatively small for performance reasons: 4 layers deep and 128 hidden units wide.

To train the multi-architecture encoding error predictor, we use the same optimizer, learning rate, and batch size as for the single-architecutre predictor. We start by freezing the weights of the pretrained classifier network, and training just the MLP regression head for 10 epochs. Then we unfreeze the classifier and train the entire network for 10 epochs, selecting the checkpoint with the epoch with the best validation accuracy.

## A.2 CHOOSING THE PSNR PREDICTOR BACKBONE ARCHITECTURE

Our single-architecture encoding error predictor is simply a fine-tuned image classification network. We used the PyTorch Image models library (TIMM) (Wightman, 2019) to quickly sweep through hundreds of candidate classification networks to fine-tune on our task. Table 3 shows the $R^2$ scores on the validation data from our initial sweep over many of the models in the TIMM library. Note that some details of training are not the same as in our final model, for example, these models were trained for fewer epochs. Table 2 shows a more thorough evaluation on some of the best models identified in Table 3. Note that for this second sweep, we excluded some of the best-performing models from Table 3 due to their slow training times. In the end, this sweep led us to choose ECA NFnet L0, a TIMM-specific variant of the NFNET architecture from Brock et al. (2021) as our final model architecture.

| Model | Validation $R^2$ | Test $R^2$ |
|---|---|---|
| eca nfnet L0 | 0.9956 | 0.9956 |
| nfnet L0 | 0.9954 | 0.9955 |
| eca nfnet L1 | 0.9952 | 0.9953 |
| coat lite mini | 0.9933 | 0.9936 |
| nf regnet b1 | 0.9932 | 0.9933 |
| convnextv2 large | 0.9923 | 0.9923 |
| coat lite small | 0.9922 | 0.9924 |

Table 2: 7 final architectures we tested before making a final decision to use nfnet for the rest of our experiments.

Table 3: 128 backbone classification architectures we tried from the Torch Image Models (TIMM) library (Wightman, 2019), on an early, smaller version of the single-architecture dataset. Ordered by the $R^2$ score they obtained on the validation set.

| model | $R^2$ | model | $R^2$ |
|---|---|---|---|
| eca_nfnet_l1 | 0.987 | eca_nfnet_l2 | 0.986 |
| convnext_large | 0.984 | convnext_base | 0.984 |
| nf_regnet_b1 | 0.983 | convnext_base_in22k | 0.981 |
| coat_lite_mini | 0.980 | convnext_tiny | 0.980 |
| convnext_xlarge_in22ft1k | 0.979 | eca_nfnet_l0 | 0.978 |
| coat_mini | 0.977 | convnext_small | 0.976 |
| nfnet_l0 | 0.975 | convnext_base_384_in22ft1k | 0.974 |
| coat_tiny | 0.973 | convnext_base_in22ft1k | 0.971 |
| deit_tiny_patch16_224 | 0.967 | crossvit_tiny_240 | 0.965 |
| crossvit_15_dagger_408 | 0.964 | coat_lite_small | 0.963 |
| crossvit_small_240 | 0.962 | crossvit_15_240 | 0.962 |
| crossvit_9_240 | 0.961 | deit_base_patch16_384 | 0.958 |
| pit_s_224 | 0.956 | crossvit_18_dagger_240 | 0.956 |
| crossvit_9_dagger_240 | 0.956 | mixer_b16_224_miil | 0.954 |
| pit_b_224 | 0.953 | crossvit_base_240 | 0.952 |
| jx_nest_base | 0.952 | jx_nest_small | 0.951 |
| cait_xxs36_224 | 0.950 | nf_resnet50 | 0.950 |
| convnext_xlarge_in22k | 0.946 | crossvit_15_dagger_240 | 0.946 |
| crossvit_18_240 | 0.945 | crossvit_18_dagger_408 | 0.944 |
| coat_lite_tiny | 0.943 | resmlp_12_distilled_224 | 0.939 |
| pit_ti_224 | 0.930 | deit_base_patch16_224 | 0.929 |
| convnext_large_in22k | 0.928 | resnetv2_50x1_bit_distilled | 0.927 |
| resnet50_gn | 0.927 | gluon_seresnext101_32x4d | 0.926 |
| cait_xxs24_224 | 0.922 | resmlp_36_distilled_224 | 0.921 |
| regnety_016 | 0.921 | regnety_006 | 0.918 |
| jx_nest_tiny | 0.916 | res2net50_14w_8s | 0.914 |
| convnext_large_in22ft1k | 0.913 | gluon_resnext101_32x4d | 0.912 |

| model | $R^2$ | model | $R^2$ |
|---|---|---|---|
| regnetz_c16 | 0.909 | regnety_002 | 0.908 |
| gluon_seresnext50_32x4d | 0.907 | gluon_senet154 | 0.905 |
| eca_botnext26ts_256 | 0.904 | ese_vovnet39b | 0.904 |
| resnest50d_4s2x40d | 0.903 | regnety_008 | 0.902 |
| regnety_040 | 0.902 | resnetv2_50x1_bitm | 0.901 |
| ecaresnet26t | 0.899 | gernet_m | 0.899 |
| resnetrs101 | 0.899 | halo2botnet50ts_256 | 0.899 |
| regnetz_d8 | 0.898 | regnetz_e8 | 0.898 |
| gluon_seresnext101_64x4d | 0.898 | deit_small_patch16_224 | 0.897 |
| regnetx_004 | 0.892 | resnetrs200 | 0.892 |
| resnetrs50 | 0.890 | convit_base | 0.889 |
| repvgg_b0 | 0.889 | eca_halonext26ts | 0.888 |
| regnetz_d32 | 0.888 | eca_resnet33ts | 0.888 |
| eca_resnext26ts | 0.887 | regnety_004 | 0.887 |
| seresnet152d | 0.885 | gcresnext26ts | 0.885 |
| resnet26 | 0.885 | regnetx_032 | 0.884 |
| resnet152d | 0.883 | seresnext26d_32x4d | 0.883 |
| pit_xs_224 | 0.883 | dpn68b | 0.881 |
| regnetx_008 | 0.881 | regnetx_016 | 0.879 |
| gluon_resnet50_v1c | 0.877 | resnet101d | 0.874 |
| resnetv2_50x1_bitm_in21k | 0.874 | resnet50d | 0.873 |
| resnetblur50 | 0.872 | cait_s24_224 | 0.872 |
| resnet26t | 0.872 | regnetx_120 | 0.871 |
| resnetv2_101x1_bitm | 0.871 | mixer_b16_224_miil_in21k | 0.870 |
| gcresnext50ts | 0.869 | lambda_resnet26t | 0.868 |
| densenetblur121d | 0.867 | regnetx_006 | 0.866 |
| gluon_resnet152_v1c | 0.866 | mixer_l16_224 | 0.865 |
| regnetx_002 | 0.865 | resnest50d_1s4x24d | 0.865 |
| gluon_resnet101_v1d | 0.865 | resnet26d | 0.865 |
| lambda_resnet50ts | 0.865 | gmlp_s16_224 | 0.864 |
| gluon_resnet50_v1d | 0.864 | bat_resnext26ts | 0.863 |
| gernet_l | 0.861 | lamhalobotnet50ts_256 | 0.861 |
| repvgg_a2 | 0.860 | repvgg_b1g4 | 0.860 |
| gluon_resnext50_32x4d | 0.858 | regnety_032 | 0.857 |
| gcresnet50t | 0.857 | resnext26ts | 0.856 |
| regnetx_080 | 0.855 | gluon_resnet101_v1c | 0.852 |
| regnety_080 | 0.850 | resnet33ts | 0.848 |

## A.3 INPUT ABLATION STUDY

Which inputs to our model are important for accurately predicting PSNR? Table 4 shows the test error of our multiple-architecture PSNR predictor with various combinations of inputs to the network removed. We see that removing any of the input features significantly degrades the accuracy of the prediction network, indicating that the final PSNR varies predictably with each of these inputs.

| Removed Input | Error ↓ | $R^2$ ↑ |
|---|---|---|
| *No inputs removed* | 0.55 | 0.987 |
| Image Size | 0.71 | 0.980 |
| $\omega_0$ | 1.25 | 0.939 |
| Number of Layers | 2.27 | 0.794 |
| Layer Size | 2.52 | 0.746 |
| All Hyperparameters | 2.90 | 0.668 |
| Image | 4.11 | 0.317 |

Table 4: Many-architecture prediction error with various removed inputs

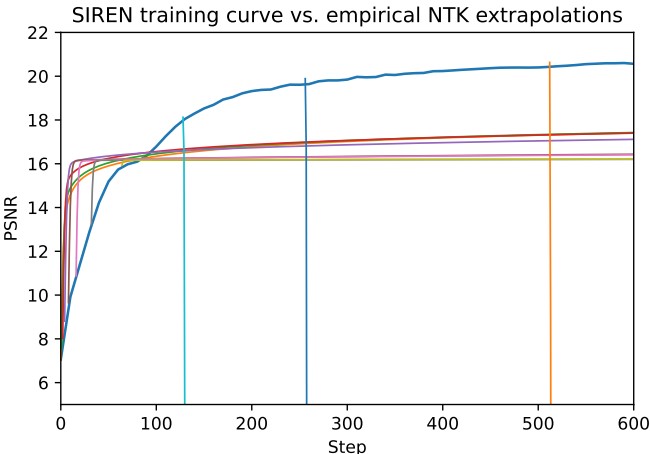

Figure 5: A typical example of a SIREN learning curve (thick blue line) vs. NTK-based extrapolations of that learning curve. When extrapolating from early epochs, the linearized network's PSNR quickly rises and then saturates. In later epochs, the large learning rate we use leads to divergent behavior (*i.e.* exploding gradients) for the linearized model, represented by the vertical lines which shoot straight down.

## A.4 EXTRAPOLATING TRAINING CURVES USING THE EMPIRICAL NTK

In some settings, gradient descent of the empirical, finite-width NTK is known to be a good and efficient approximation for optimizing the neural network itself (Mohamadi et al., 2023). Zancato et al. (2020) used this approximation to successfully predict the training times on fine-tuning tasks to within 20% accuracy. With full-batch training, the MSE loss $\mathcal{L}$ of the NTK approximation after $t$ steps of training with a learning rate $\eta$ develops according to the following equation (Arora et al., 2019):

$$\mathcal{L}(y_t) = ||(I - \eta\Theta)^t(y - y_0)||_2^2 \tag{5}$$

where $\Theta$ is the NTK matrix, $y_0$ is the network's output at initialization, and $y$ is the target output.

Unfortunately, we did not find success in applying this approximation to our networks. Following the method from Zancato et al. (2020), we approximate the empirical NTK of the network at initialization and used this to extrapolate the network's learning curve. For our problem, these NTK-based extrapolations saturate very early, predicting that the network will essentially learn a flat-colored image where each pixel just takes on the average color of the whole image.

Others have found that even when the NTK at initialization is a poor proxy for network convergence, the empirical NTK can become a significantly better approximation after just a few steps of training (Kopitkov & Indelman, 2020). To test if this held for our problem, we took snapshots of our SIRENs' weights after $n = 1, 2, 4, 8..., 2^{15}$ training steps, and used the NTK approximation from Zancato et al. (2020) to extrapolate the learning curves forward from that point onward.

Figure 5 shows the true training curve vs. empirical-NTK-extrapolated training curves starting from $n = 1, 2, 4, 8..., 2^{15}$ training steps. It is clear that the SIRENs and their NTKs behave very differently.

At first, the NTKs produced from snapshots of the network early in training follow the exact same dynamic described above: they learn only the DC component of the target function, and then saturate.

Around 128 training steps, the learning rate of $\eta = 0.002$ leads to divergent behavior on the NTK model. This is due to the increasing eigenvalues of the kernel matrix $\Theta$: according to Equation 5,

SGD will begin to diverge when the largest eigenvalue of K exceeds $\frac{1}{\eta}$. Instead of decreasing, the loss increases exponentially with each training step.

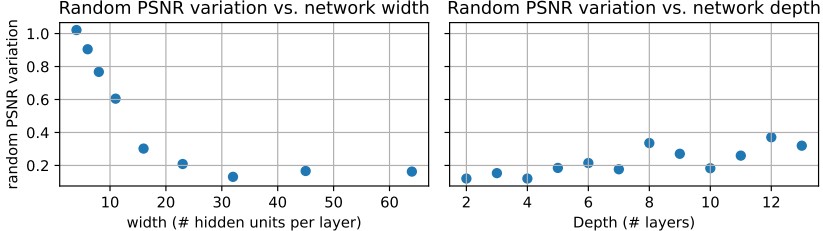

Figure 7: Effect of SIREN width and depth on the random variation in PSNR. Both sweeps begin from a baseline architecture of 10 layers, 32 hidden units per layer. In the width sweep, the depth is held constant, and vice-versa. For each choice of width/height, 10 SIRENs are trained on the same image, and the standard deviation in PSNR is plotted. Notice how the random variation in PSNR increases dramatically for narrow networks.

## A.5 RANDOM VARIATION IN SIREN PSNR (CONTD.)

Despite comprising less than 1% of the network's learnable parameters, the first layer is responsible for roughly 66.0% of COIN's variance in final PSNR due to random initialization. By initializing the first layer in the same way each time instead of randomly, we reduce the standard deviation in PSNR among SIRENs using Dupont et al.'s method from 0.18 to 0.10 PSNR.

This random variation is inversely correlated with network width (See Figure 7). When SIRENs are wide, they take larger random samples of the space of random Fourier features, and these large random samples converge to a similar distribution. But for very narrow net-

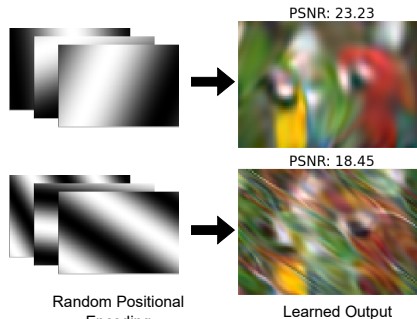

Figure 6: Output of two small SIRENs with randomly-initialized 3-channel positional encodings.

works, such as the ones used by COIN, the positional encoding consists of only a few randomly sampled sine waves, so there is significant variation in the quality of the positional encoding. Figure 6 illustrates this principle for very small SIRENs with a width of just 3 hidden units in the first layer. In this extreme case, the effect of a poorly initialized PE layer is especially clear.

We can improve SIREN performance by selecting a good first layer, instead of choosing one at random. We train 100 randomly initialized COIN networks on each of the 24 Kodak images, for a total of 2,400 trained networks. For each of the 24 images, we take the first layer from the SIREN with the best final PSNR, and train 24 new SIRENs, one for each image in the Kodak dataset, which are initialized with this layer. This gives us a collection of $24 \times 24 = 576$ SIRENs whose first layers are not randomly initialized, but have been selected as the best out of 100 randomly initialized layers. In the interest of training time, we reduce the number of training steps for each of the 2400 + 576 SIRENs in this experiment from 50,000 to 20,000.

When retraining on the same image using the positional encoding with the best PSNR, our SIRENs perform 0.23 PSNR better on average than with random PEs. When applying the best PE from one image to another image, average improvement in PSNR is 0.14 PSNR. This suggests that good positional encodings are somewhat transferable from image to image. Figure 8 shows the results of our experiment in detail.

### A.5.1 Explanation of Equation 2

We would like to estimate the irreducible error of our encoding error prediction problem. If we are trying to predict a variable Y from a variable X, the minimum possible RMSE that any regression model can reach is:

$$RMSE = \sqrt{E[Var[Y|X]]}$$

In our setting, Y is the encoding error of a SIREN as measured in PSNR, and X represents the SIREN's training inputs: width, depth, $\omega_0$ and the target image. $Y|X = x$ is the distribution of encoding errors a given SIREN setup $x$ will reach across different random seeds.

To estimate $E[Var[Y|X]]$, we randomly sample $N$ SIREN training inputs $x_1, \ldots, x_N \sim X$ from our dataset, and estimate the variance $S_i^2 \approx Var[Y|X = x_i]$ for each. This gives us the following irreducible error formula:

$$RMSE \approx \sqrt{\frac{1}{N} \sum_{i=1}^{N} S_i^2} \tag{6}$$

We can estimate the variance $S^2$ of a population from $m$ samples using the unbiased sample variance estimator:

$$S^2 = \frac{1}{m-1} \sum_{i=1}^{m} (E[y] - y_i)^2 \tag{7}$$

To estimate $Var[Y|X = x_i]$, we draw two samples $y_{i,1}, y_{i,2} \sim Var[Y|X = x_i]$. So $m = 2$, and $E[y] = \frac{y_{i,1} + y_{i,2}}{2}$. By substituting and simplifying, Equation 7 becomes:

$$S_i^2 = \frac{1}{2}(y_{i,1} - y_{i,2})^2 \tag{8}$$

Finally, we can substitute Equation 8 into Equation 6 to get Equation 2.

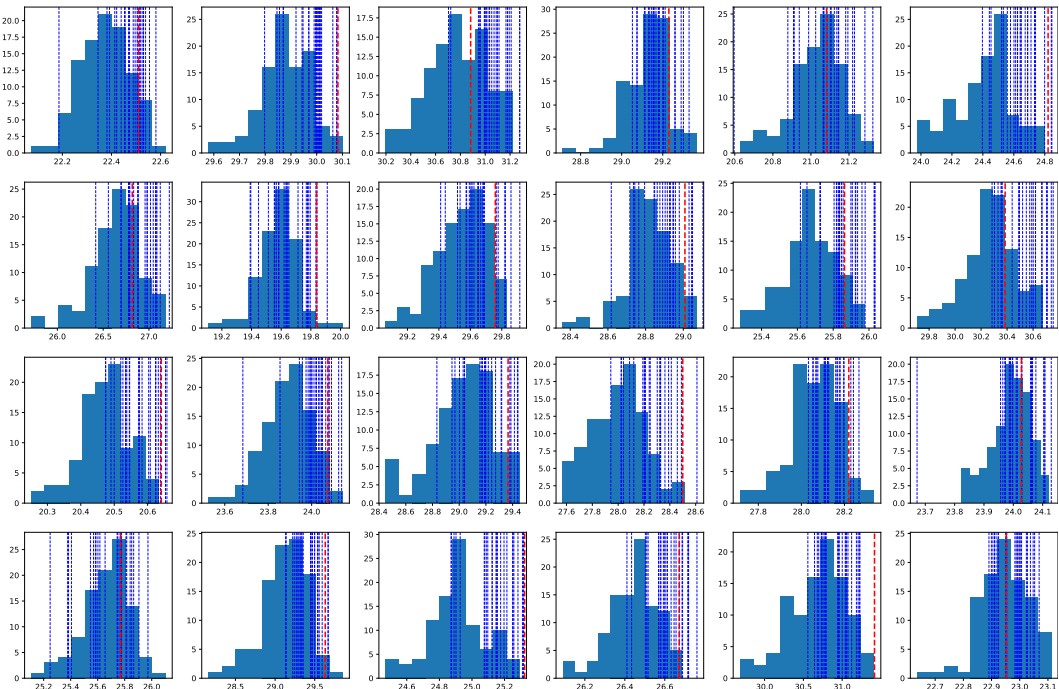

Figure 8: Histograms of the PSNR which the default COIN network reaches on each of the 24 images in the Kodak dataset across 100 different random seeds. For each image, we pick the randomized run with the best PSNR, extract its first layer, and reuse that first layer in an otherwise newly randomized network on each image. Each vertical line represents the PSNR reached by one of these retrained networks that used the best first layer out of 100. The blue vertical lines indicate that the first layer was selected from a network trained on a different image, the red vertical lines indicate the models that were retrained on the same image.

## A.6 SIRENS COMPRESS THE GLOBAL COLOR SPACE

Figure 9 illustrates the significant qualitative differences in the information lost by JPEG and COIN compression. At the same compression ratio, COIN better maintains the sharpness of high-contrast edges. Artifacts and distortions in COINs appear more structurally complex. Most interestingly, COINS appear to globally compress an image's color space; reducing variation in color to reduce the amount of information per pixel. We posit that this global color compression accounts for much of COIN's advantage over JPEG at low bitrates.

If our hypothesis is correct, then our JPEG proxy models for SIREN encoding error (from Figures 2a and 2b) could be further improved by accounting for the discrepancy in how COINs and JPEGs compress color space. But this remains a topic for future work.

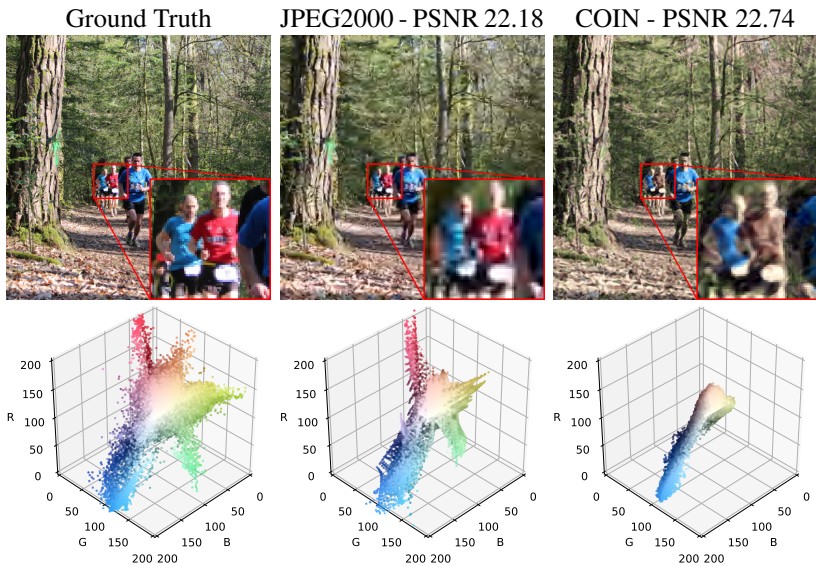

Figure 9: Images and their pixel-color scatter plots in RGB space. Note how the SIREN sacrifices color fidelity for spatial fidelity to achieve a higher net PSNR than JPEG2000 at 0.3 bpp.

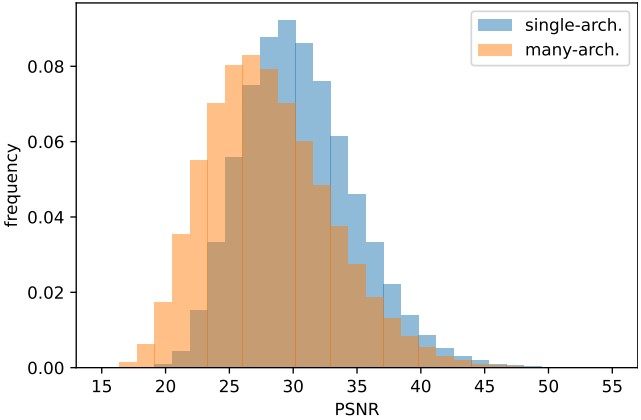

Figure 10: PSNR distribution for the single- and many-architecture SIREN datasets.