# OpenReview forum: "Predicting the Encoding Error of Implicit Neural Representations"
_ICLR.cc/2024/Conference — ICLR 2024 Conference Withdrawn Submission_

### Official Review · Reviewer_Dqyn · 2023-10-16

**Soundness:** 3 good
**Presentation:** 4 excellent
**Contribution:** 2 fair
**Rating:** 5
**Confidence:** 3

**Summary:**

This paper proposes a method for predicting the PSNR of a SIREN fit to a photograph. The authors constructs a large dataset of SIRENs with randomly sampled hyperparameters, fitted to various photographs. The image is passed through a pre-trained CNN to extract features, while the SIREN hyperparameters are fed through a positional encoding. An MLP is trained on these inputs to predict the PSNR of the SIREN with the given hyperparameters fit to the image. The authors discover that the random initialization of the SIREN can have a noticeable effect on the PSNR. They also find that JPEG compression correlates strongly with the SIREN's fitting success (COIN compression), with JPEG PSNR explaining 98% of the variance in the SIREN PSNR.

**Strengths:**

The authors are very thorough in their experiments, carefully probing many hyperparameters with a very large dataset. This makes the paper fairly unique within the topic of INRs. This scientific rigor is indeed rare in the field as a whole. The paper also makes some interesting observations such as the dependence of the SIREN performance on random initialization which is reduced for wider networks, and the relationship between JPEG compression and SIREN quality.

**Weaknesses:**

In opting for a narrow but carefully controlled study, the paper has unfortunately limited its potential impact. The authors seem to acknowledge that fitting SIREN to photographs is not a task of significant practical interest, but they claim that studying this setting can inform real applications of INRs in the way that studying fruit flies informs human biology. The problem with this analogy is that unlike in biology, we have no idea how much overlap there is between the behavior of SIREN and the behavior of NeRF or other common INRs, and my guess is that if you take any modern INR parameterization (say, ZipNeRF or Gaussian splatting), there is very little overlap in their fitting behaviors. Even just in the context of 2D images, SIREN may not be the best parameterization. Why not investigate other types of INRs like Instant NGP which can scale much better to large images? Indeed I think if the authors undertook a similarly careful investigation of the hash table vs. MLP behaviors of Instant NGP in this same 2D setting, that would make for a paper with much stronger relevance and impact. More ambitiously, I would love to see experiments that show a similar approach can be used to estimate the encoding error of NeRFs or SDFs (e.g. fit to Objaverse XL to achieve a similar large scale dataset). As is, I think the paper would have important takeaways for only a somewhat small subset of ICLR's audience (primarily those interested in COIN compression), which does not include most INR/NeRF practitioners.

**Questions:**

See weaknesses

---

### Official Review · Reviewer_bVWr · 2023-10-30

**Soundness:** 3 good
**Presentation:** 3 good
**Contribution:** 2 fair
**Rating:** 3
**Confidence:** 4

**Summary:**

This paper studies the problem of predicting the encoding error that an INR will reach on a given training signal. The authors developed a dataset of 300,000 SIREN samples, including 100,000 SIRENs with the same architectural hyperparameters and 200,000 varying-size SIRENs. A predictor is built in this paper to predict the encoding error of a given SIREN structure, with an intuitive regression module to predict the PSNR value. This paper also contributes to many experimental analyses, including:
(1) the influence of random initializations to SIREN convergence (Section-4);
(2) the correlation between the fitting quality of SIREN and the compression quality of JPEG (Section-5). Concretely, they have a linear correlation across images, by comparing the PSNR of SIREN with the PSNR of JPEG at fixed bitrates;
(3) predicting the SIREN encoding error (Section-6), like the "power laws" of the fitting quality of SIREN against the number of model parameters.

**Strengths:**

It is appreciated that the authors spent a lot of resources (roughly 12000 GPU hours) building a dataset that contains 300,000 converged SIREN models. It is known that the common implicit neural representations (INRs) take thousands of iterations to learn the network parameters. Therefore, it may be useful if we have the ground truth of appropriate SIREN models corresponding to the original signals (images in this paper).

Also, in this paper, the experimental analyses of the encoding error of SIREN are clear, probably guiding the implementations of INRs in the field of compression to some extent. For example, previously it is known that the random initializations of SIREN models influence the fitting quality, now it is more clear how much such influence would be.

**Weaknesses:**

Although there are very detailed experimental analyses provided in this paper, this paper lacks non-trivial results. From my opinion, all the experimental results in this paper are unsurprising or very intuitive. Personally, the experimental section in this paper is more like a technical report. For example, the relationship between SIREN and JPEG representations can be inferred even though we have not conducted this experiment before: if we set the bitrate as a constraint, an image with more complex context should be encoded by JPEG with lower PSNR value. At the same time, the SIREN model will surely fit this image with lower PSNR.

Besides, the dataset contribution in this paper, i.e., 300,000 pretrained SIREN models, is hard to be considered as a significant one.
(1) On the one hand, it is very simple to run a script thousands of times to fit the SIREN models to represent different images.
(2) On the other hand, if we consider in depth, we already know the initializations indeed take some impact to the final reconstruction quality, why not build a dataset with SIREN models whose parameters are already aligned so that it would be easier to capture the redundancies across network parameters? If the dataset itself is built with randon initializations, it would be very hard to build a generative model / VAE-style compression model to further capture the redundancies across network parameters.
In other word, it may be more useful to build a dataset with some disciplines, like in VC-INR [Ref1, missed in this paper], the network parameters are controlled by low-rank matrices so that a VAE-style compression model can be trained upon the dataset of network parameters (a set of network parameters correspond to an image). Or like in COMBINER [Ref2, cited as Guo et al. in this paper], the network parameters are trained with the constraint of rate-distortion tradeoff. In the scenarios of [Ref1] or [Ref2], it would be more useful to build the dataset of model parameters, because the dataset itself is already aligned to easily learn a generative model of INRs.


[Ref1] Modality-Agnostic Variational Compression of Implicit Neural Representations. Schwarz et al., ICML2023.
[Ref2] Compression with Bayesian Implicit Neural Representations. Guo et al., NeurIPS 2023.

**Questions:**

Except the abovementioned points, could the author try to elaborate more about how such a dataset with 300,000 SIREN models would inspire some specific problems/applications in the future?

---

### Official Review · Reviewer_w62A · 2023-10-31

**Soundness:** 2 fair
**Presentation:** 2 fair
**Contribution:** 2 fair
**Rating:** 3
**Confidence:** 3

**Summary:**

This paper presents a method which predicts the encoding loss that a popular INR network (SIREN) will reach, given its network hyperparameters and the signal to encode.

**Strengths:**

1. The idea of "predicting the model performance" could be interesting.

**Weaknesses:**

1. The paper's structure is somewhat disorganized, as the authors appear to conflate the methods and experiment sections. This might make it challenging for readers to understand the proposed ideas and their corresponding experimental support. I recommend the authors revise the paper's organization to adhere to the traditional structure, with distinct sections for methods and experiments.

2. The proposed method lacks scalability, as it appears to be restricted to a specific set of candidate architectures. While the authors' efforts are commendable, the approach may become ineffective when a new architecture, particularly one with significant design changes, is introduced. The authors should note that this dataset is entirely dependent on SIREN, and consider how they might adapt their method to more diverse architectures.

3. The role of JPEG in this study is unclear to me. The results presented in Figures 2(a) and 2(b) seem rather straightforward. For instance, Figure 2(a) essentially provides a zoomed-in view of the rate-distortion (RD) curve shown in Figure 2 of the COIN paper. This COIN figure indicates that the JPEG RD and COIN RD intersect at approximately 0.3 bpp. The authors could provide more context or interpretation to make these results more meaningful.

4. Regarding JPEG, the authors state: "find the encoding error of a SIREN architecture on a few images, and then use a standard zeroth-order numerical optimizer (e.g., scipy.optimize) to find the corresponding JPEG compression ratio which maximizes the correlation between JPEG and SIREN encoding error." If I understand correctly, this implies that the derived JPEG compression ratio could be used to compress new images, with the PSNR of these new images serving as a reference for the corresponding SIREN PSNR. If this is the case, why not simply use the proposed neural network-based error predictor? The authors should clarify whether there is a performance difference between these two approaches.

**Questions:**

see comments above.

---

### Official Review · Reviewer_Ny7t · 2023-11-04

**Soundness:** 1 poor
**Presentation:** 2 fair
**Contribution:** 2 fair
**Rating:** 3
**Confidence:** 4

**Summary:**

This paper studies how one can build a nice predictor of a PSNR level that a SIREN may achieve after training for a fixed number of training steps with some learning rate, with the given set of architectural hyperparameters. To do this, authors first build a large-scale dataset of SIRENs, measuring the final PSNR after training under various settings. Then, authors try several options to train a good predictor for the final PSNR, including a model that predicts based on JPEG error, a model based on fitting the scaling law, and a neural network predictor. Authors conclude that NFNet-based predictor works favorably.

**Strengths:**

- **Computational effort.** It is indeed a significant computational effort to construct the dataset described in the paper. Although it is not very clear at this point exactly how much of a detail will be available in the released dataset, it will nevertheless be, at least, a mild help in doing the neural fields research.

- **Clarity.** The sentences are mostly clearly written and the paper is generally easy to comprehend.

**Weaknesses:**

- **Measuring the final PSNR.** I do not think measuring the final PSNR provides an effective measure. In many cases, the training PSNR of a SIREN is a very unstable object (as the authors acknowledge). The PSNR curve tends to increase first, but suddenly they drop significantly. For this reason, it is quite typical and more practical to use the "model that achieves the best PSNR until k-th step" instead of the final model. In fact, this is what is plotted in COIN, which is also cited in this paper (see Figure 4). If the authors are not already doing this (correct me if the authors are already doing this!), predicting the max PSNR may be a more practically meaningful task.
- **Obscure goal.** It is difficult to tell what practical impact the work (and the dataset) may have---it would be much better if the paper is more explicit about this point. I presume that the main goal is to free ourselves of the need to tune the SIREN hyperparameters for each new image. If so, I believe that authors could have directly compared the mean PSNR reached, by (1) following a fixed default set of hyperparameters, and by (2) using the predicted-to-be-optimal hyperparameters.
- **Limited Practical Impact (minor).** The present work mainly considers the task of 2d image regression with SIREN. 2D regression is not the most promising application of neural fields, and SIREN is already a bit old architecture---there has been much progress, and I cannot think of a good reason why one would prefer to use SIREN over other options. Thus I believe that the practical impact of the present work, per se, may not be quite big. Nevertheless, the developed technique may be able to be adapted to other tasks and models (one may need to check this point, however), so I think this subject is still of academic interest.
- **Clarity: Formalism.** Some key results are buried in the text and hard to locate. For instance, I believe that the the results in section 6.3 and 6.4 need to be more clearly delivered to the readers, in the form of a table or a figure.

**Questions:**

- **Generalization.** I wonder how a predictor trained for a specific type of workloads (i.e., the dataset) generalize to another type of workloads. For instance, do they generalize to medical datasets, e.g., CheXpert?